# Diagnostic Strategies for Treatment Selection in Advanced Prostate Cancer

**DOI:** 10.3390/diagnostics11020345

**Published:** 2021-02-19

**Authors:** Ciara S. McNevin, Anne-Marie Baird, Ray McDermott, Stephen P. Finn

**Affiliations:** 1Department of Histopathology and Morbid Anatomy, Trinity Translational Medicine Institute, Trinity College Dublin, D08 W9RT Dublin, Ireland; MCNEVINC@tcd.ie; 2Department of Medical Oncology, St. James Hospital, D08 NHY1 Dublin, Ireland; 3School of Medicine, Trinity Translational Medicine Institute, Trinity College Dublin, D02 A440 Dublin, Ireland; BAIRDA@tcd.ie; 4Department of Medical Oncology, Tallaght University Hospital, D24 NR0A Dublin, Ireland; RAY.MCDERMOTT@tuh.ie; 5Department of Medical Oncology, St. Vincent’s University Hospital, D04 YN26 Dublin, Ireland; 6Department of Histopathology, St. James’s Hospital, P.O. Box 580, James’s Street, D08 X4RX Dublin, Ireland

**Keywords:** prostate cancer, treatment strategies, castration-resistant prostate cancer, CRPC, systemic therapy, dynamic classification, metastatic hormone-sensitive prostate cancer, mHSPC

## Abstract

Prostate Cancer (PCa) is a leading cause of morbidity and mortality among men worldwide. For most men with PCa, their disease will follow an indolent course. However, advanced PCa is associated with poor outcomes. There has been an advent of new therapeutic options with proven efficacy for advanced PCa in the last decade which has improved survival outcomes for men with this disease. Despite this, advanced PCa continues to be associated with a high rate of death. There is a lack of strong evidence guiding the timing and sequence of these novel treatment strategies. This paper focuses on a review of the strategies for diagnostic and the current evidence available for treatment selection in advanced PCa.

## 1. Introduction

PCa is the second most common cancer diagnosis made in men and the fifth leading cause of death worldwide [1]. Advanced age is the principal risk factor with more than 75% of all PCa diagnosed in men aged 65 years and older [2]. The worldwide PCa burden is expected to grow to almost 2.3 million new cases and 740,000 deaths by 2040 simply due to the growth and aging of the population [3] making it an expanding global health concern. In addition to age, ethnicity (e.g., African American), heredity (e.g., family history) and mutations in DNA-repair genes such as BRCA2 are risk factors for PCa [4,5,6]. A variety of environmental factors have been discussed as being associated with the risk of developing PCa [4]. Japanese men for instance have a lower PCa risk (incidence of 2/100,000) compared to men from USA (14/100,000). Yet, when Japanese men move from Japan to California, their risk of PCa increases, approaching that of American men, implying a role of environmental or dietary factors [7,8].

For most men with PCa, their disease will follow an indolent course. The 5-year survival rates are encouraging; 98% and 83% in the USA and Europe, respectively [1]. Localized PCa may be cured with surgery or radiation therapy, and unlike other cancers there can be a role for active surveillance. However, the disease recurs in approximately 20 to 30% of men treated for localized PCa and advanced disease is associated with poor outcomes. Although most men with metastatic PCa initially respond to Androgen-Deprivation Therapy (ADT), the median duration of sensitivity to ADT in metastatic castration-sensitive prostate cancer (mCSPC) ranges from between 24 and 36 months. Inevitably their cancer progresses on this treatment to a disease state known as castration-resistant prostate cancer (CRPC). The median survival for men with metastatic castration-resistant prostate cancer (mCRPC) is less than 2 years [9]. Despite an advent of new therapies in the last decade, PCa remains the second leading cause of cancer mortality among males. The treatment landscape for PCa is increasingly rich with options, yet therapeutic selection has never been so complex. This can lead to treatment selection dilemmas for clinicians which can impact outcomes if not carefully considered. This review paper focuses on a review of the current strategies for diagnostic and treatment selection and sequence in advanced PCa.

## 2. Advancement in Diagnostic Strategies Aiding Selection of Treatment in Metastatic Prostate Cancer

### 2.1. Screening Biomarkers in Prostate Cancer

There are several biomarkers identified for PCa screening [10,11] (Table 1) with the most ubiquitous and widely adopted being the Prostate-Specific Antigen (PSA), which was approved by the US Food and Drug Administration (FDA) in 1986. PSA screening has had a large impact on trends in PCa incidence and mortality and its widespread adoption has been scrutinized since its discovery 40 years ago [12]. Although PSA screening appeared highly beneficial, there is a question of whether it was improving survival or simply detecting earlier and possibly insignificant PCa, resulting in over treatment. Recent data from the US Preventative Services Task Force (USPSTF) reported evidence that PSA screening offers a potential benefit of reducing the chance of death from PCa in men aged 55–69 years [13]; however, it does not recommend screening over the age of 70 years. The European Association of Urology (EAU) recommends PSA screening for men over 50 years of age or 45 if they have a family history. They recently added that men carrying BRCA2 mutations > 40 years of age should be screened [5]. PSA is also widely used as a tool in active surveillance and to assess response to treatment. Additional biomarkers have developed recently for example Prostate Health Index (PHI) has been approved by the FDA. The PHI is a diagnostic blood test that combines free and total PSA and the (−2) pro-PSA isoform (p2PSA). The PHI test is intended to reduce the number of unnecessary prostate biopsies in PSA-tested men [14]. Similarly, Prostate Cancer Antigen 3 [PCA3) is a known urinary biomarker for prostate cancer. It is a non-coding, prostate-specific mRNA that is highly over-expressed in 95% of PCa cells. The non-invasive nature of the urine PCA3 test makes the PCA3 test attractive for clinicians, giving it superiority over several alternative biomarkers with similar or higher specificity [15], The 4Kscore showed superiority over PSA in diagnosing indolent versus aggressive PCa [16]. Despite these new advances, at present the PSA remains the international gold standard for screening and surveillance.

### 2.2. Histopathological Grading of PC

Pathological grading of prostate cancer has been established since the 1970′s with the development of the eponymously named Gleason Grading System. It was born from a prospective, randomized study looking into clinicopathological correlation of over 2900 patients with PC [17]. The Gleason Score (GS) is the result of the sum of the most common and the highest grades in biopsy material and the most common and second most common patterns in radical prostatectomies. Since its development, the GS has adopted several modifications, the most recent of which was proposed and adopted by the World Health Organisation (WHO) in 2016 after International Society of Urological Pathology (ISUP) consensus. The new classification provided more accurate stratification of tumors with the aim to simplify the number of grading categories and reducing overtreatment of indolent PC [18]. Offerman et al. recently published a systemic review which concluded the prognostic accuracy of the ISUP 2014/WHO 2016 grade groups [19]. Recently a grade group system has been introduced to simplify the score interpretation for patients and clinicians by removing scores attributed in the past as Gleason 1 and/or 2. Grade group 1 consists of GS 6 or lower (low-grade cancer); grade group 2 consists of GS 3 + 4 = 7 (medium-grade cancer); grade group 3 consists of GS 4 + 3 = 7 (medium-grade cancer); grade group 4 consists of GS 8 (high-grade cancer) and grade group 5 consists of GS 9 to 10 (high-grade cancer) [18]. The Gleason system remains an essential prognostic tool in contemporary management of PC. Its prominence has led it to being used as a correlation metric to assess new diagnostic and prognostic strategies [10,20,21,22].

**Table 1 diagnostics-11-00345-t001:** Screening and Predictive Biomarkers in Prostate Cancer.

Biomarker	Marker Type	Specimen	Utility	FDA Approved (Year)
Prostate-Specific Antigen (PSA)	Protein	Blood	Initial Bx/Surveillance	Yes (1986) [23]
Prostate Health Index (PHI)	Protein	Blood	Initial Bx/Surveillance	Yes (2012) [14]
4kscore Test	Protein	Blood	Initial Bx/Surveillance	No [16]
Prostate Cancer Antigen 3 [PCA3)	mRNA	Urine	Repeat Bx	Yes [24]
Select MDx (DLX1/HOX6)	mRN	Urine	Initial Bx	No
Mi-Prostate Score (TMPRSS2-ERG)	mRN	Urine	Initial Bx	No
ExoDx	mRNA	Urine	Initial Bx	No
ConfirmMDx (Epigenetics Panal)	Protein	Tissue	Repeat Bx	No

### 2.3. Imaging Techniques for Advanced Prostate Cancer

Defining the extent of PCa spread in men with newly diagnosed or with recurrent PCa is crucial for therapeutic decision making. Historically Computed Tomography (CT) and radionuclide bone scintigraphy (BS) were the imaging modality of choice; however newer, more sensitive imaging methods are improving detection of cancer spread. Magnetic Resonance Imaging (MRI) combined with dynamic contrast-enhanced MRI showed a higher sensitivity and specificity compared with MRI alone in detecting local recurrences after radical prostatectomy [25]. Whole Body MRI showed to have greater sensitivity and specificity for detecting bone metastases when compared with conventional imaging techniques such as CT and BS [26]. A recently published study compared conventional primary staging of PCa with Prostate-Specific Membrane Antigen (PSMA) positron emission tomography PET/CT. The authors reported that PSMA PET-CT demonstrated greater accuracy compared to conventional imaging [27]. Findings of this study support the replacement of conventional imagine by PSMA PET/CT in primary staging of PCa however prospective analyses are needed to confirm a possible beneficial effect on survival outcomes of these newer imaging techniques.

### 2.4. Prognostic Biomarkers

In an era of personalized medicine the need for genetic prognostic markers is critical and a precision medicine framework for mCRPC has been lagging compared to other cancers. Clinicians use several clinical markers in predicting prognosis in PCa intuitively such as metastatic status (i.e., number of bone/visceral metastases), performance status (PS), time to CRPC, and number of prior treatments. PSA, Gleason Score, Lactate Dehydrogenase, Alkaline phosphatase, Albumin, Hemoglobin, Neutrophil-lymphocyte ratio and testosterone, all of which have proven prognostic or predictive properties in advanced PCa [28]. However, these indicators are crude and fail to guide treatment selection with any specificity. Recently, the identification of baseline circulating tumor cell (CTC) count at the start of treatment for mCSPC was highly prognostic of 7-month PSA response and of PFS at 2 years in a phase III trial of mCSPC [29]. The authors hypothesized that baseline CTC count may serve as a valuable prognostic marker to discriminate men likely to respond favorably to hormonal therapies from those who may benefit from early alternate interventions.

#### 2.4.1. DNA-Repair Mutations

Genomic instability is a hallmark of cancer and genetic alterations have become topical in PCa owing to scientific advancements that have expanded understanding in this area. Historically, unselected primary tumor tissue obtained at radical prostatectomy was the main source of material for genomic profiling of prostate cancer and DNA defects were thought to be uncommon. For example, germline mutations in BRCA2 were underappreciated as a driver of hereditary prostate cancer as only 1–3% of unselected localized diagnoses harbor BRCA2 germline mutations [30]. However, it has been reported that the percentage of patients with germline mutations in DNA-repair genes increases with advancing disease with 4.6% of men identified with germline mutations in localized disease and 11.8% to 16.2% observed in metastatic disease [31]. Pritchard et al. reported observing incidence rates of germline genetic mutations in men with metastatic PCa as follows; BRCA2 (5.35%); CHEK2 (1.87%), ATM (1.59%) and BRCA1 (0.87%). BRCA2 was reported to have the highest relative risk of metastases when compared with men who do not have PCa [31]. DNA damage repair defects (DDR) represent 25% of these alterations, with BRCA2 mutations being the most frequently observed [6]. A second analysis by Dall’Era et al. looked at both somatic and germline DNA alterations in men with both localized and metastatic disease. They found that 154 men from a cohort of 944 (16.3%) with PCa harbored a mutation, the most frequent being BRCA2 and ATM [32].

The presence of these mutations has prognostic, predictive, and therapeutic value for men with PCa. It can also have family implications if a germline mutation is identified. Germline BRCA2 mutations are associated with higher risk of PCa development, risk of local treatment failures and mortality [6]. More importantly, these alterations are clinically actionable and can be therapeutically targeted with Poly (adenosine diphosphate ribose) polymerase (PARP) inhibition or by immune checkpoint inhibition. Currently there are two PARP inhibitors, Olaparib and Rucaparib, approved by the FDA which have shown survival benefit in patients with BRCA1, BRCA2, or ATM mutations [33,34] (Figure 1) (Table 2). The cohort studied had responded to PARP inhibition after progression of disease while receiving a novel Androgen-Receptor Signaling Inhibitor (ARSi) such as Abiraterone and Enzalutamide. Therefore, these therapies present new treatment lines for men with otherwise few options left to them [35]. (See Section 3.2.7). Pembrolizumab was recently approved by the FDA for unresectable or metastatic, microsatellite instability-high (MSI-H) or mismatch repair deficient (dMMR) solid tumors that have progressed following prior treatment which includes PCa with MSI-H or dMMR [29]. It is noteworthy that most of the studies analyzing the presence of DDR defects in tumors do not distinguish the germline or somatic origin of the variants identified [36]. Ongoing research evaluating the predictive impact of DNA-repair mutations combined with greater ease of access to genetic testing for patients, will no doubt expand clinical knowledge for clinicians on this topic.

#### 2.4.2. Distorted Androgen-Receptor Signaling and Genetic Alterations

In CRPC androgen-receptor signaling is distorted in a variety of ways due to genetic alterations. It is now well established that CRPC continues to rely on AR signaling. This has led to the development of novel ARSi’s in CRPC [38,39]. A detailed discussion of genetic alterations in advanced PCa is beyond the scope of this review however (i) AR Promiscuity, (ii) AR Amplification, (iii) Modification of Co Regulators, (iv) Aberrant Activation and (v) AR Variants have all been identified as AR resistance mechanisms to ADT [40]. The androgen-receptor splice variant 7 messenger RNA (AR-V7), results in the truncation of the ligand-binding domain, and has recently come into prominence. Its presence has a strong association with Enzalutamide and Abiraterone resistance [41]. AR-V7 protein expression was observed as rare in primary tumors (<1%) but common in metastatic tumors (75%), suggesting that AR-V7 expression adaptively increases under the selective pressure of AR-directed therapies [41]. Clinically validated assays are now commercially available for the AR-V7 biomarker; however, a large-scale prospective trial is needed for validation of clinical utility.

## 3. Treatment Selection for Advanced Prostate Cancer

### 3.1. Drugs with Proven Efficacy in Metastatic Castration-Sensitive Prostate Cancer (MCSPC)

#### 3.1.1. Androgen-Deprivation Therapy for mCSPC

It is now well established that the AR signaling pathway is a major driver of PCa growth and ADT has been the standard of care for relapsed PCa since the seminal studies by Huggins in the last century for which he received a Nobel Prize [42,43,44]. ADT was first line therapy for men with newly diagnosed advanced PCa or mCSPC for over 70 years. ADT is initially effective in metastatic PCa with 95% of men showing a response; however, all men inevitably experience progression to CRPC [45]. Adverse effects reported with ADT include hot flashes, sexual dysfunction, metabolic syndrome, anemia, and skeletal and cardiac morbidity [46]. Once men developed mCRPC they historically had few treatment options, and the median survival for these men is <2 years [9].

#### 3.1.2. Docetaxel for mCSPC

Docetaxel has efficacy in the treatment of mCRPC, which was demonstrated in two large phase three Randomized Control Trials (RCTs) in 2004 namely Tax327 and SWOG9916 [47,48] (Table 3). Therefore, it was rational to investigate the efficacy of Docetaxel combined with ADT as 1^st^ line treatment for newly diagnosed mCSPC. Docetaxel is a chemotherapy drug that works by binding to microtubules and preventing microtubular polymerization resulting in cell cycle arrest. It has also been shown to inhibit AR activity and its resultant downstream pathways [49]. The hypothesis was it could potentially target androgen independent clones that may be inherently resistant to ADT [50]. In 2015 three large RCTs; CHAARTED [51], STAMPEDE [52] and GETUG-15 [53] (Table 3) evaluated ADT combined with 6 cycles of Docetaxel and demonstrated a survival benefit, with Hazard ratios of 0.61 (0.47–0.80 95% CI); 0.76 (0.62–0.93 95% CI) and 0.88 (0.68–1.14 CI), respectively. 

GETUG-15 reported an absolute difference in median overall survival (OS) of 14 months; however, the benefit did not reach statistical significance (62.1 vs. 48.6 months; hazard ratio (HR): 0.88; 95% confidence interval [CI]: 0.68–1.14; *p* = 0.3) and the trial was negative [53]. Of note, the progression-free and radiologic progression-free survival were significantly longer in the ADT plus Docetaxel arm. CHAARTED demonstrated a statistically significant median OS benefit of 13.6 months with the addition of Docetaxel (57.6 months vs. 44.0 months; HR: 0.61; 95% CI: 0.47–0.80; *p* < 0.001). The benefit was more prominent in the subgroup with high-volume disease than the overall study population (49.2 vs. 32.2 months; HR: 0.60; 95% CI: 0.45–0.81; *p* < 0.001). STAMPEDE, the largest of the three trials, and demonstrated that the addition of Docetaxel was significant in OS (77 vs. 67 months; HR: 0.76; 95% CI: 0.63–0.91; *p* = 0.003) and in failure-free survival (37 vs. 21 months; HR: 0.62; 95% CI: 0.54–0.70; *p* < 1 × 10^−10^) in the overall study population. Of note, the addition of zoledronic acid (a bisphosphonate) with Docetaxel did not confer any survival benefit.

A meta-analysis of the use of all available data from the RTC’s comparing Docetaxel with the standard of care in CSPC showed that the addition of Docetaxel to standard of care improved survival (HR for death of 0.77 (95% CI: 0.68–0.87; *p* < 0.0001)) [67]. According to the authors, this translates to an absolute improvement in 4-year survival of 9% (95% CI: 5–14). Upfront ADT and 6 cycles of Docetaxel is now licensed as first line treatment for mCSPC.

#### 3.1.3. Abiraterone for mCSPC

Abiraterone is a potent, selective, and irreversible inhibitor of CYP17, which inhibits androgen synthesis [68] and was the first of the novel androgen-receptor signaling inhibitors (ARSi). Similar to Docetaxel, it was iterative to explore if Abiraterone had efficacy in mCSPC given it had demonstrated survival benefit in men with mCRPC. The STAMPEDE and Latitude trials evaluated this [37,57,58] (Table 3).

In the Latitude study men were randomly assigned to ADT, Abiraterone acetate and prednisone, or ADT alone. All patients were high-risk, mCSPC, and were chemo-naïve. The overall rate of survival at 3 years was 66% in the Abiraterone group and 49% in the placebo group with a HR for death of 0.62; 95% (CI), 0.51 to 0.76; *p* < 0.001) demonstrated a clear survival benefit. The STAMPEDE trial enrolled men who had PCa that was newly diagnosed and metastatic, node-positive, or high-risk locally advanced (i.e., included men with both metastatic and non-metastatic disease). Similar to the Latitude study they randomized men to receive ADT alone or with Abiraterone and prednisolone. There was strong evidence of a survival advantage in the treatment combination group, with a 3-year survival of 83% as compared with 76% in the ADT-alone group (HR for death, 0.63; 95% confidence interval (CI), 0.52 to 0.76; *p* < 0.001).

A systematic review and meta-analysis reported a 14% absolute improvement in OS at 3 years through the addition of Abiraterone and prednisolone to ADT (SOC), from 55% to 69%. The authors reported that the results across the two trials were remarkably consistent, and there was no evidence of statistical heterogeneity [69]. The analysis also evaluated that secondary outcome of HR to the PFS from Latitude and STAMPEDE translates to a 28% absolute improvement in PFS at 3 years with addition of Abiraterone and prednisolone to ADT, from 30% to 58%. Abiraterone primary toxic events in these studies were hypertension, hypokalemia, and elevated hepatic enzymes. The meta-analysis reported an approximate three-fold increase in grade III–IV acute cardiac (Peto Odds Ratio (OR) = 2.93, 95% CI 1.74–4.93, *p* < 0.001) and hepatic toxicity (Peto OR = 3.09, 95% CI 2.12–4.50, *p* < 0.001) and an approximate two-fold increase in grade III–IV vascular events (OR = 2.28, 95% CI 1.71–3.03, *p* < 0.001), the majority of which (≥90%) were related to hypertension. Together these studies provided the basis for adding Abiraterone and prednisolone to ADT as SOC for men with metastatic disease at diagnosis however Abiraterone should be used with caution in men with underlying cardiac morbidity [70].

#### 3.1.4. Androgen-Receptor Targets for mCSPC

Enzalutamide is the second of the new androgen-receptor signaling inhibitors (ARSi) developed in the last decade. Enzalutamide it is an oral targeted AR inhibitor that competitively binds to the ligand-binding domain of the AR and inhibits AR translocation to the cell nucleus. It also blocks the recruitment of AR cofactors, and AR binding to DNA and induces apoptosis [39,71]. Enzalutamide was approved by the FDA for mCSPC off the back of the Arches Trial which demonstrated efficacy in improving survival [72]. The trial enrolled 1150 men with mCSPC randomized (1:1) to receive either Enzalutamide or placebo. All men received some form of ADT. The risk of radiographic progression or death was significantly reduced with Enzalutamide plus ADT vs. placebo plus ADT (HR, 0.39; 95% CI, 0.30 to 0.50; *p* < 0.001; median not reached vs. 19.0 months). There was also a statistically significant improvement reported in the Enzalutamide arm compared to placebo in time to initiation of a new antineoplastic therapy (HR 0.28; 95% CI: 0.20, 0.40; *p* < 0.0001).

Apalutamide is an oral nonsteroidal antiandrogen agent that binds directly to the ligand-binding domain of the androgen receptor and prevents androgen-receptor translocation, DNA binding, and androgen-receptor–mediated transcription. In the TITAN trial, men with mCSPC were randomly assigned in a 1:1 ratio to receive Apalutamide or matched placebo in addition to ADT. The OS percentage at 24 months was 82.4% in the Apalutamide group and 73.5% in the placebo group (HR for death, 0.67; 95% CI, 0.51 to 0.89; *p* = 0.005) [65].

#### 3.1.5. Therapeutic Sequence in mCSPC

There remains a need for prospective RTCs comparing Docetaxel, Abiraterone, Enzalutamideand Apalutamide in mCSPC. Sydes et al. conducted a direct, randomized comparative analysis of ADT plus Abiraterone versus ADT plus Docetaxel in mCSPC and reported similar cancer-specific survival, symptomatic skeletal events and adverse events concluding that the efficacy of the two treatments in prolonging survival appear to be equivalent. [73].

Further Analyses from the CHAARTED and GETUG-AFU15 studies reported that the survival benefits associated with Docetaxel early during treatment, particularly in men with high-volume metastatic disease, are substantially larger than the survival benefits associated with using Docetaxel later after castration resistance has developed [74]. However, how it compares when combined with each of the newer anti androgen therapies in this population remains unknown. The recent approval of Enzalutamide and Apalutamide for mCSPC by the FDA has added yet more complexity to the selection of treatment options. RCTs with direct head-to-head comparison of therapies would provide us with valuable granularity on the optimum sequence of therapies. An obvious starting point would be the comparison of first line treatment options, followed by an exploration of the most efficacious second-line therapies. Of note, selection of men for treatment based on molecular biomarkers has been notably absent in studies of hormone treatment for patients with an initial diagnosis of metastatic disease.

### 3.2. Drugs with Proven Efficacy in Metastatic Castration-Resistant Prostate Cancer (MCRPC)

#### 3.2.1. Docetaxel for mCRPC

Docetaxel was the first of many significant advances made in the treatment of PCa in the last 15 years. As referenced above, two large phase three RCTs published in 2004 investigated Docetaxel as first line treatment in mCRPC compared to the standard of care (Mitoxantrone) [Table 3]. The first of these TAX327 reported the median duration of survival was 18.9 months (95% CI, 17.0 to 21.2) in the group given Docetaxel every 3 weeks compared to 16.5 months (95% CI, interval, 14.4 to 18.6) in the mitoxantrone group. The HR for death reported was 0.83 (95% CI, 0.70 to 0.99; *p* = 0.04). Almost simultaneously the SWOG9916 reported a median survival of 17.5 months among the men assigned to Docetaxel and estramustine and 15.6 months among the men assigned to mitoxantrone and prednisone (*p* = 0.02); the corresponding HR for death was 0.80 (95% CI, 0.67 to 0.97). This advancement was the first therapy to show survival benefit in mCRPC and thus was greeted positively. However, it excludes a significant portion of men who are not be eligible for Docetaxel due to poor PS, pre-existing medical conditions, or tolerability concerns. Therefore, there was an impetus to develop therapies that could be administered *in lieu* of chemotherapy.

#### 3.2.2. Abiraterone for mCRPC

Abiraterone has shown efficacy when used both post and pre-Docetaxel in mCRPC as shown in two RCTs; COU-AA 301 (15.8 vs. 11.2 months; HR: 0.74; 95% CI: 0.54–0.70; *p* < 1 × 10^−10^) [57,58] and as reported in the final analysis of COU-AA 302 [39,60] (34.7 vs 30.3 months; HR: 0.81; 95% CI, 0.70 to 0.93; *p* = 0.0033), respectively. Prior the COU-AA-302 trial, there was no validated level I evidence showing that treatments other than chemotherapy had value in the chemotherapy-naïve mCRPC population [75]. Abiraterone can avoid chemotherapy related toxicity and provides a treatment option for men who may be ineligible for Docetaxel. The most frequent side effects of Abiraterone experienced were mineralo-corticoid associated side effects such as hypertension, hypokalemia, and oedema. Since Abiraterone is administered with prednisolone, men are also at risk of long-term side effects to steroid exposure. Of note, the COU-AA-302 trial demonstrated that the addition of Abiraterone to prednisone delayed the development and progression of pain. The trial reported a 10-month difference in the median time to opiate use. In the final analysis, the median time to opiate use for PCa related pain was 33.4 months in the Abiraterone acetate plus prednisone group vs. 23.4 months in the prednisone plus placebo group (HR = 0.72, *p* < 0.0001) [75].

#### 3.2.3. Androgen-Receptor Targets for mCRPC

Similar to Abiraterone, Enzalutamide has shown efficacy when used both pre- and post-Docetaxel in mCRPC as shown in two RCTs. The AFFIRM trial reported a median OS of 18.4 months in the Enzalutamide group versus 13.6 months in the placebo group with a HR of 0.63 (95% [CI] 0.53, 0.75; *p*  <  0.001) and [59]. Based on these results, it was recommended that the study be halted and unblinded, with eligible men in the placebo group offered treatment with Enzalutamide. Patients were randomly assigned to receive either oral Enzalutamide or placebo. Presence of visceral disease was permitted; however previous chemotherapy exposure was not. The study reported that 72% in the Enzalutamide group, as compared with 63% in the placebo group, were alive at the data-cutoff date (HR for death, 0.71; 95% CI, 0.60 to 0.84; *p* < 0.001), providing another treatment option with proven survival benefit for men with mCRPC.

Of note, the AR inhibitors Apalutamide, Darolutamide and Enzalutamide have recently been approved for the treatment of non-metastatic CRPC based on phase 3 trials showing significantly longer metastasis-free survival with these agents than with placebo [38,66,76].

#### 3.2.4. Cabazitaxel for mCRPC

In 2010 FDA approved Cabazitaxel as a new therapeutic option for men with mCRPC resistant to Docetaxel. Cabazitaxel is a novel tubulin-binding taxane with poor affinity for P-glycoprotein [54]. The TROPIC trial investigated Cabazitaxel in men with mCRPC who had disease progression during or after treatment with Docetaxel. OS was 15.1mths (95% CI 14.1–16.3) in the Cabazitaxel group and 12.7mths (11.6–13.7) in the mitoxantrone group. The HR for death of men treated with Cabazitaxel compared with those taking mitoxantrone was 0.70 (95% CI 0.59–0.83, *p* < 0.0001). This solidified Cabazitaxel as second-line chemotherapy option for men who had progression after Docetaxel. Docetaxel and Cabazitaxel were compared head to head as first line chemotherapy for men with mCRPC however this trial failed to show that Cabazitaxel was superior to Docetaxel in men with mCRPC who had not received previous chemotherapy [77].

The efficacy of Cabazitaxel compared to the ARSi’s in men with mCRPC who were previously treated with Docetaxel and had progression while receiving the alternative inhibitor (Abiraterone or Enzalutamide) were unclear. De Wit [78] investigated this in a recent RCT. The median PFS was 8.0mths in the Cabazitaxel group, as compared with 3.7mths in the androgen-signaling–targeted inhibitor group. HR for imaging-based progression or death, 0.54; 95% CI, 0.40 to 0.73; *p* < 0.001). Cabazitaxel resulted in a risk of death from any cause that was 36% lower than that with Abiraterone or Enzalutamide, despite 33% of the men in the ARSi group crossing over to receive Cabazitaxel at the time of progression. Multiple trials have shown survival benefit with the use of several therapies, but only one large trial to date has compared two life-prolonging agents—Docetaxel and Cabazitaxel. This trial failed to show that Cabazitaxel was superior to Docetaxel in men with mCRPC who had not received previous chemotherapy [77].

#### 3.2.5. Immunotherapy for mCRPC

Sipuleucel-T, an autologous cellular immunological agent, became was approved by the FDA in 2010 as a result of the IMPACT trial, which was a double-blind, placebo-controlled, multicenter trial which assigned 341 men to receive sipuleucel-T and 171 assigned to receive placebo. The median OS was 25.8 months for men receiving sipuleucel-T and 21.7 months for controls (HR for death in the sipuleucel-T group, 0.78; 95% CI, 0.61 to 0.98; *p* = 0.03). It is approved in both the pre- and post-Docetaxel setting however expense and logistical convenience limits its use in practice [79]. Pembrolizumab, an antibody that targets programmed death 1 (PD-1), has proven efficacy in cancers with defective mismatch repair cells, which occur in 5 to 12% of men with mCRPC [31]. Early phase I and II studies assessed the antitumor activity and safety of pembrolizumab in cohorts of mCRPC patients with promising results however further phase III trials are required [29,37].

#### 3.2.6. Radium 223 for mCRPC

Radium-223 is an alpha-particle–emitting radionuclide that binds preferentially to the hydroxyapatite in osteoblastic bone metastases. A phase 3 trial showed that radium-223 resulted in improved OS as compared with the best standard of care alone, for men who had symptomatic bone metastases without visceral metastasis. OS was prolonged in the radium-223 group (median 14.9 months, vs. 11.3 months in the control group; HR for death, 0.70; 95% CI, 0.58 to 0.83; *p* < 0.001) [61]. Men who had not previously received chemotherapy and those previously treated with Docetaxel both had improved OS with radium-223 as compared with placebo. A phase III RTC investigated the addition of radium-223 to Abiraterone acetate plus prednisone however this combination did not improve symptomatic skeletal event-free survival in men with mCRPC and was associated with an increased frequency of bone fractures compared with placebo therefore it is not recommended as a treatment combination [80].

#### 3.2.7. Molecular Targets for mCRPC

Olaparib is a PARP inhibitor with known efficacy in ovarian and breast cancer for patients with an identified DNA damage repair gene. PROfound was a phase III RTC comparing Olaparib with an ARSi in two cohorts of men with mCRPC. Eligible patients were men (≥18 years of age) with confirmed mCRPC whose disease had progressed during treatment with Enzalutamide or Abiraterone. Previous taxane chemotherapy was allowed [33].

Cohort A (patients with BRCA1/BRCA2 or ATM gene alterations) were randomized to receive Olaparib or receive either Abiraterone or Enzalutamide. In Cohort B patients with 12 other pre-specified DNA Damage repair genes alterations were randomized to receive Olaparib or hormonal treatment. In the overall population, the PFS survival was significantly longer in the Olaparib group than in the control group (5.8 m vs. 3.5 m; HR, 0.49; 95% CI, 0.38 to 0.63; *p* < 0.001). This benefit was more pronounced in Cohort A. 7.4 m vs. 3.6 m; HR for progression or death, 0.34; 95% (CI), 0.25 to 0.47; *p* < 0.001). The FDA subsequently approved Olaparib for adult patients with deleterious or suspected deleterious germline or somatic homologous recombination repair (HRR) gene-mutated mCRPC, who have progressed following prior treatment with Enzalutamide or Abiraterone. The final analysis of this study reported the median duration of OS was 17.3 months with Olaparib and 14.0 months with control therapy (HR for death, 0.79; 95% CI, 0.61 to 1.03) [35].

TRITON2 investigated the efficacy of Rucaparib for men with deleterious BRCA mutation (germline and/or somatic)-associated mCRPC who have been treated with androgen-receptor-directed therapy and a taxane-based chemotherapy. The results of this study have led to its recent approved by the FDA [34].

#### 3.2.8. Therapeutic Sequence in mCRPC

Currently in men without an identifiable MSI-H, dMMR or DNA-repair mutation high level data support the use of sipuleucel-T, Enzalutamide, Abiraterone, Docetaxel, and radium-223 in selected populations of men with mCRPC as a first line therapy as depicted in Figure 1. Grade A evidence also supports the use of Enzalutamide, Abiraterone, Cabazitaxel, and radium-223 in selected patients after treatment with Docetaxel. Historically mCRPC treatments have been categorized into pre- and post-Docetaxel groups simple due to the chronology of drug development, which is artificial and suboptimal [81]. Furthermore, there is a paucity of prospectively validated phase III head-to-head comparison trials and clinicians are often reliant on small retrospective analyses to guide decisions around sequencing treatments [82].

High level data that is available to support treatment sequencing selection in mCRPC include the phase III trial, FIRSTANA in which Cabazitaxel failed to show superiority over Docetaxel in men with mCRPC who had not received previous chemotherapy [77]. Docetaxel remains the first line chemotherapy choice in practice for this cohort. A phase III trial also demonstrated Cabazitaxel significantly improved several clinical outcomes such as OS and image-based progression-free survival, as compared ARSi in men mCRPC who had been previously treated with Docetaxel and the ARSi (Abiraterone or enzalutamide) [78]. The current questions that remain are (i) is Docetaxel or an ARSi superior as first line therapy in mCRPC and (ii) if ARSi is the superior choice, which of the two, Abiraterone or Enzalutamide, is more efficacious? In addition, further granularity is needed on the most suitable patient population, potential cross-resistance mechanisms, optimal sequential dosing, and possible combination strategies relating to these treatment options.

A systemic review by Zhang et al. evaluated the indirect comparisons of Abiraterone and Enzalutamide in retrospective studies and reported similar survival benefits with both agents in men with mCRPC before and after chemotherapy. It was noted, however, that Enzalutamide may be advantageous for secondary endpoints including time to PSA progression, radiographic PFS, PSA response rate, time to quality-of-life deterioration, and time to initiation of chemotherapy [83]. Maines et al. also performed a systematic review and descriptive analysis to explore the clinical outcomes of mCRPC patients who were treated with third-line ARSi after having previously received Docetaxel and another ARSi. The results of their analysis suggest that Enzalutamide before Abiraterone may lead to a slightly longer OS than the reverse sequence, and more significantly, that the use of Cabazitaxel and an ARSi in any order seems to offer an even greater OS advantage [84].

It is likely that as science continues to advance, biomarkers will direct clinical treatment selection. This is already being realized with the discovery of AR variants associated with resistance to ARSi’s [85]. These findings require large-scale prospective validation before they can be of clinical utility. Precision medicine is moving closer in the management of mCRPC with the approval of Pembrolizamab by the FDA for people with MSI-H and /or dMMR and PARP inhibitors for people with DNA-repair damage mutations. Although these pockets of progress are positive for subgroups of patients, it does not negate the need for large head-to-head comparative studies for first- and second-line treatments for mCRPC.

## 4. Conclusions

PCa is an important public health issue which is only likely to become more prevalent as global demographics evolve. Clinical progress for treating metastatic PCa in the past decade has been remarkable. There has been leaps in drug development for advanced PCa among various drug classes, with recent approvals for second generation taxanes, molecular targeted therapies, multiple targeted androgen-receptor drugs, and immunotherapies. Although this juggernaut of progress is positive for the field, the therapeutic landscape has never been so complex. There is paucity of high-grade evidence to guide clinicians in treatment selection and treatment sequence. Furthermore, PCa is a heterogenous disease, yet historically treatments have not been based on molecular stratification and biological diversity. A role for biomarkers to select men that may benefit from a particular therapy will need to be elucidated further, but the detection of the AR-V7 splice variant and DNA-repair mutations appear promising candidates in the quest for biomarkers that will allow the precision medicine revolution to take place.

## Figures and Tables

**Figure 1 diagnostics-11-00345-f001:**
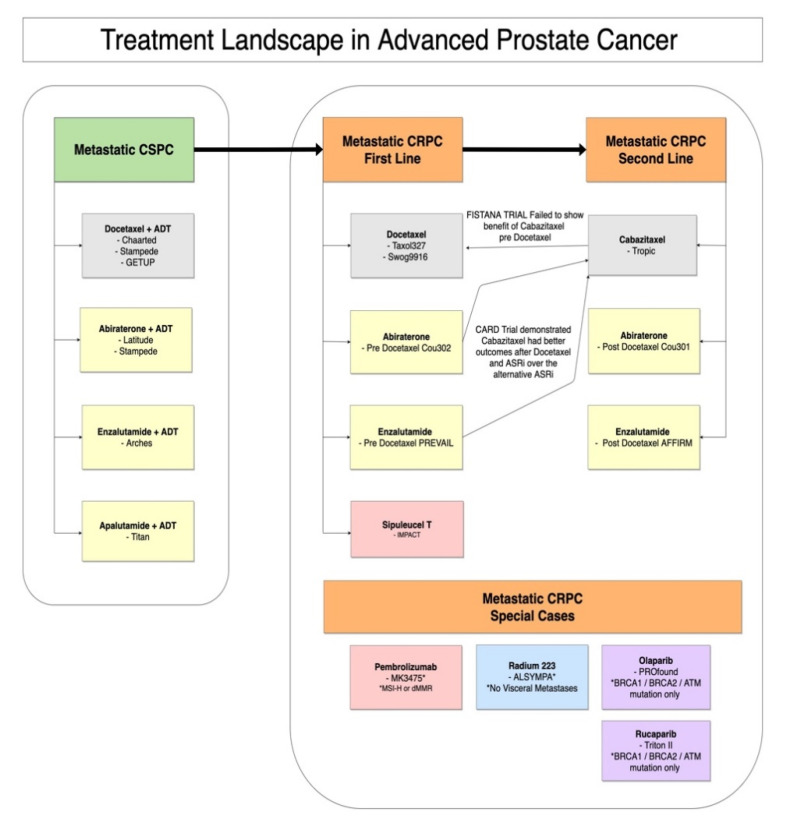
Treatment Landscape for advanced PC.

**Table 2 diagnostics-11-00345-t002:** DNA Damage Repair (DDR) Pathways in Prostate Cancer.

DNA Damage Repair Pathways in Prostate Cancer	Repairs	Clinical Actionable
Homologous recombination repair	Double Stranded Breaks	Olaparib/Rucaparib [33,34]
non- homologous end joining Non-homologous end joining	Double Stranded Breaks	Olaparib/Rucaparib [33,34]
Mismatch Repair Base Excision Repair	Base Mismatches Single Stranded Breaks	Pembrolizmab [37]
Nucleotide Excision Repair	Inter strand Crosslinks	
Translesion Synthesis	Bulky Adducts	

**Table 3 diagnostics-11-00345-t003:** Practice Changing Trials for Treatment for Metastatic Prostate Cancer.

Year	Trial	Study Treatment	Control	(*n*)	Pretreated Chemo (c) ADT (h) (ARSi)	Visceral Disease Allowed on Trial	Indication	Sequence Approved by FDA	HR for Death (95% CI)	Author
2004	Tax 327	Docetaxel	Mito + P	1006	(h)	Yes	mCRPC	1st line	0.76 (0.62–0.94)	Tannock [47]
	SWOG 9916	Docetaxel + Estramustine	Mito + P	674	(h)(c)	Yes	mCRPC	1st line	0.80 (0.67–0.97)	Petrylak [48]
2010	TROPIC2	Cabazataxel	Mito + P	755	(h)(c)	Yes	mCRCP	2nd line	0.70 (0.59–0.83)	de Bono [54]
	IMPACT	Sipuleucel-T	Placebo	512	(h)(c)	Yes	mCRPC	1st line	0.77 (0.61–0.98)	Kantoff [55]
2011	NCT00321620	Denosumab	ZA	1904	(h)	n/a	mCRPC	1st line	1.03 (0.91–1.17)	Fizazi [56]
	COU-AA-301	Abiraterone + P	Placebo + P	1195	(h)	No	mCRPC	2nd line	0.65 (0.54–0.83)	de Bono [57,58]
2012	AFFIRM	Enzalutamide	Placebo	1199	(h)(c)	Yes	mCRPC	2nd line	0.63 (0.53–0.75)	Scher [59]
2013	COU-AA-302	Abiraterone + P	Placebo + P	1088	(h)	No	mCRPC	1st line	0.75 (0.61–0.93)	Ryan [60]
	ALSYMPCA	Radium223 + SOC	SOC	921	(h)	No	mCRPC	1st line	0.70 (0.58–0.83)	Parker [61]
2014	PREVAIL	Enzalutamide	Placebo	1717	(h)	Yes	mCRPC	1st line	0.71 (0.60–0.84)	Beer [62]
2015	CHAARTED	Docetaxel + SOC	SOC	790	*	Yes	mCSPC	1st line	0.61 (0.47–0.80)	Sweeney [51]
	STAMPEDE	Docetaxel + SOC	SOC	917	**	Yes	mCSPC	1st line	0.76 (0.62–0.92)	James [52]
	GETUG-15	Docetaxel + SOC	SOC	385	***	Yes	mCSPC	1st line	0.88 (0.68–1.14)	Gravis [53]
2017	LATTITUDE	Abiraterone + P + SOC	Placebo + P	1199	****	Yes ^	mCSPC	1st line	0.62 (0.51–0.76)	Fizazi [63]
	STAMPEDE	Abiraterone + P + SOC	Placebo + P	1917	*****	Yes	mCSPC	1st line	0.61 (0.49–0.75)	James [64]
2019	TITAN	Apalutamide	Placebo	1207	(h)(c)	Yes	Post-Doc	2nd line	0.67 (0.51–0.89)	Chi [65]
	ARAMIS	Darolutamide	Placebo	1509	(h)	No	CRPC	1st line	0.71 (0.50–0.99)	Fizazi [66]
2020	PROFOUND	Olaparib	Placebo	387	(h)(c)(ARSi)	Yes	mCRPC^B^	2nd line	0.55 (0.29–1.06)	De Bono [33]
	TRITON2	Rucaparib	Placebo	115	(h)(c)(ARSi)	Yes	mCRPC^B^	3rd line	N/A	Abida [34]

Mito denotes Mitoxantrone; P denotes Prednisolone; ZA denotes Zolendronic Acid; SOC denotes Standard of Care; CRPC denotes Non-Metastatic Castration-Resistant Prostate Cancer; * Prior adjuvant ADT was allowed if the duration of therapy was 24 months or less and progression had occurred more than 12 months after completion of therapy. Patients who were receiving ADT for metastatic disease were eligible if there was no evidence of progress; ** All patients were intended for long-term hormone therapy, started no longer than 12 weeks before randomization; *** ADT for patients with metastatic disease could have been initiated no more than 2 months before enrolment; **** Patients were allowed 3 months or less of androgen-deprivation therapy with luteinizing hormone–releasing hormone analogues; ^ Visceral Disease was Mandatory for Enrolment; ***** <12 months of total ADT with an interval of >12 months without treatment). Patients were intended for treatment with long-term ADT that started no longer than 12 weeks before randomization; mCRPC^B^ denotes patients with CRPC harboring a mutation in BRCA1/2 with mCRPC. N/A Not Applicable—HR for death was not a distinct end point and therefore cannot be reported.

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
