# Peer review of "Diagnostic Strategies for Treatment Selection in Advanced Prostate Cancer"

_diagnostics, 2021, doi:10.3390/diagnostics11020345_

Round 1

Reviewer 1 Report

This review by the authors summarizes diagnostic strategies for treatment selection in advanced prostate cancer. The topic is generally interesting, and the summarized progress is useful in the field of treatment in advanced prostate cancer. The manuscript is also very well-written. Altogether, the manuscript is worth publishing for researcher in the area of advanced prostate cancer after the authors address the comments below.

Specific comments:

The only suggestion I have is that as a review article, it would be nice to include one or more pictures describing the therapeutic sequence in mCSPC and mCRPC.

Author Response

The only suggestion I have is that as a review article, it would be nice to include one or more pictures describing the therapeutic sequence in mCSPC and mCRPC.

Thank you for this suggestion. We have now included a picture depicting the therapeutic sequence in mCSPC and mCRPC.  

Reviewer 2 Report

A very well written and comprehensive paper.

Please could you include micro fluids, PCA3 in the diagnosis section.

Once done the paper is good to be accepted. 

Author Response

A very well written and comprehensive paper.

Please could you include micro fluids, PCA3 in the diagnosis section.

Once done the paper is good to be accepted. 

Thank you for your review and this comment. On the first draft PCA3 was included in Table One as ‘Prostate Cancer Antigen 3’ and has now been expanded upon in further detail in text from Section 2.1 (Screening Biomarkers for Prostate Cancer).  

Reviewer 3 Report

This work by McNevin et al. is a very well written paper with an exhaustive review on the issue of prostate cancer, one of the know big killer disease.

In this Reviewer's opinion Authors just should emphasize the role of histology on patient management, as two off the Authors are pathologists.

My only concern is about the journal target: this review paper could be fitting in other journals, as the diagnostic section is limited, while the therapeutical options are more detailed, as stated by the difference of subheadings fo the two sections.

Author Response

This work by McNevin et al. is a very well written paper with an exhaustive review on the issue of prostate cancer, one of the know big killer disease.

In this Reviewer's opinion Authors just should emphasize the role of histology on patient management, as two off the Authors are pathologists.

Thank you for noting this limitation. We have now included description and brief history of the evolution of the histological grading in an additional subsection which was added (2.2. Histopathological Grading of PC) which explains the Gleason Staging system and modifications thereof.

My only concern is about the journal target: this review paper could be fitting in other journals, as the diagnostic section is limited, while the therapeutical options are more detailed, as stated by the difference of subheadings fo the two sections.

We agree with this comment. This has been addressed above with the inclusion of an additional subsection of histopathological grading of Prostate Cancer and expansion of the role of PCA3.

Round 2

Reviewer 3 Report

The Authors have revised the paper accordingly to the Reviewers' comments fulfilling their suggestions.